https://doi.org/10.1038/s41467-020-14927-4　　**OPEN**

# Structural evolution at the oxidative and reductive limits in the first electrochemical cycle of $Li_{1.2}Ni_{0.13}Mn_{0.54}Co_{0.13}O_2$

Wei Yin [1,2], Alexis Grimaud [1,3], Gwenaelle Rousse[1,2,3], Artem M. Abakumov[4], Anatoliy Senyshyn[5], Leiting Zhang[6], Sigita Trabesinger [6], Antonella Iadecola[3], Dominique Foix[7], Domitille Giaume[8] & Jean-Marie Tarascon [1,2,3 ✉]

High-energy-density lithium-rich materials are of significant interest for advanced lithium-ion batteries, provided that several roadblocks, such as voltage fade and poor energy efficiency are removed. However, this remains challenging as their functioning mechanisms during first cycle are not fully understood. Here we enlarge the cycling potential window for $Li_{1.2}Ni_{0.13}Mn_{0.54}Co_{0.13}O_2$ electrode, identifying novel structural evolution mechanism involving a structurally-densified single-phase A' formed under harsh oxidizing conditions throughout the crystallites and not only at the surface, in contrast to previous beliefs. We also recover a majority of first-cycle capacity loss by applying a constant-voltage step on discharge. Using highly reducing conditions we obtain additional capacity via a new low-potential P" phase, which is involved into triggering oxygen redox on charge. Altogether, these results provide deeper insights into the structural-composition evolution of $Li_{1.2}Ni_{0.13}Mn_{0.54}Co_{0.13}O_2$ and will help to find measures to cure voltage fade and improve energy efficiency in this class of material.

[1] Chimie du Solide et de l'Energie, UMR 8260, Collège de France, 75231 Paris Cedex 05, France. [2] Sorbonne Université, 4 Place Jussieu, 75005 Paris, France. [3] Réseau sur le Stockage Electrochimique de l'Energie (RS2E), CNRS FR 3459, 33 Rue Saint Leu, 80039 Amiens, France. [4] Center for Energy Science and Technology, Skolkovo Institute of Science and Technology, 3 Nobel Street, Moscow 143026, Russia. [5] Forschungsneutronenquelle Heinz Maier-Leibnitz (FRM II), Technische Universität München, Lichtenbergstrasse 1, 85748 Garching, Germany. [6] Electrochemistry Laboratory, Paul Scherrer Institute, Forschungsstrasse 111, 5232 Villigen PSI, Switzerland. [7] IPREM - UMR 5254 CNRS, Université de Pau et des Pays de l'Adour, Hélioparc, Avenue Pierre Angot, 64053 Pau Cedex 9, France. [8] Chimie ParisTech, PSL University, CNRS, Institut de Recherche de Chimie Paris, 75005 Paris, France. ✉email: jean-marie.tarascon@college-de-france.fr

ithium (Li) ion batteries have doubled their energy density since early commercialization in 1990s[1]. New boost might be given by the emergence of Li-rich layered oxide, whose archetypical composition $Li_{1.2}Ni_{0.13}Mn_{0.54}Co_{0.13}O_2$ (Li-rich NMC) delivers capacities as high as 300 mAh g$^{-1}$ and where part of the transition metals is replaced by Li[2,3]. The first electrochemical cycle of Li-rich NMCs show a peculiar two-step charge profile, followed by a sloping S-shaped discharge curve, in which the cationic redox activity could only account for around half of the discharge capacity. Various hypotheses have been proposed to explain the extra capacity enlisting transition-metal over-oxidation[4,5], irreversible oxygen loss with surface densification[6,7], Li$^+$/H$^+$ exchange[8], $Li_2O$ removal with "$MnO_2$-like" activation[9], oxygen release/re-accommodation[10], oxygen redox at the interphase[11], and so on.

However, now it is well accepted, based on our early works on model compounds ($Li_2Ru_{1-y}Sn_yO_3$ series) and other complementary experimental[12–14] and theoretical results[15–17], that the extraordinary capacity of Li-rich layered oxides is due to the cumulative contribution of both cationic and anionic reversible redox processes. This led not only to better understanding and the discovery of new $nd$-metal-based Li-rich layered oxides[18,19] but also to the emergence of a new class of cathodes called "Li-rich cation-disordered oxides"[20,21]. In spite of the large capacity offered by these Li-rich cathodes and of intense research efforts, their commercialization has remained unsuccessful. This is because the extra capacity offered by anionic redox generally comes with practical drawbacks, such as voltage fade, large voltage hysteresis, and poor rate capability[22]. Although quite remarkable progress has been made in understanding these Li-rich materials, we are still far from finding the practical measures needed for their commercial success. Thus, the need persists for revisiting Li-rich NMCs with a fresh perspective on a few remaining questions that have not yet reached a full consensus, thus hindering their industrial deployment.

Specifically, one of the questions is "what are the mechanistic details of the staircase versus S-shaped evolution during the first cycle?" To address this question, the feasibility of eliminating structural disorder and restoring cation ordering by simply heating a discharged sample, as reported by Singer et al.[23], is of paramount importance since it highlights the role of disorder. Going deeper into this question, Yu et al.[24] have recently showed the appearance of a core-shell structure within particles at potentials >4.4 V using X-ray fingerprint associated with an additional Bragg peak next to the main-phase (0 0 3) one. This peak was proposed to be associated with a disordered spinel structure, growing at the surface of the particle, which reversibly converts back to rocksalt structure upon lithiation. However, neither dynamics of this conversion upon cycling was studied nor any mean to promote it was proposed. The second question, which naturally arises, is "why do Li-rich NMCs display larger first-cycle irreversibility (nearly ~15%) as compared to the classical ones?" Irreversible loss of oxygen was earlier proposed as the main cause[6,25,26] and was recently complemented by the feasibility of having a not fully reversible oxygen redox process[13,27]. However, a legitimate yet not rationalized question, whether the oxygen evolution accounts for all the irreversible capacity, still remains unanswered.

Thus, we decided to revisit the electrochemical–structural relationships in $Li_{1.2}Ni_{0.13}Mn_{0.54}Co_{0.13}O_2$ by exploring deeper the behavior of this compound under severe oxidative-reductive conditions. In this study, for the first time, we were able to experimentally stabilize, under oxidizing conditions, a densified single-phase material, enlisting Mn migration into the octahedral interlayer sites. Moreover, we demonstrated that the anionic redox process triggers an electrochemical activity at low potentials, which has never been reported before, accounting for a part of the first-cycle irreversible capacity loss.

## Results

**Lattice densification on deep oxidation.** A pure $Li_{1.2}Ni_{0.13}Mn_{0.54}Co_{0.13}O_2$ phase, as evidenced by the Rietveld refinement of synchrotron X-ray diffraction (XRD) in Supplementary Fig. 1 and Supplementary Table 1, was prepared by combined co-precipitation and solid-state synthesis[28]. Electrochemical tests are made in Swagelok cells versus Li. As typically reported for this material, the two-step charge profile converts into a sloping S-shaped one on discharge[28]. Upon cycling, the charge and discharge profiles nearly superimpose, which are however tainted by capacity decay and voltage fade (Supplementary Fig. 2).

As a first step, we explored the structural evolution during the first charge under various oxidizing conditions by recording operando X-ray powder diffraction patterns for $Li_{1.2}Ni_{0.13}Mn_{0.54}Co_{0.13}O_2$/Li cells (Supplementary Fig. 3). After charging to 4.8 V, in accordance with previous reports[13,29,30], a shoulder peak starts to appear at the high-angle side of the original main-phase (0 0 3) peak (denoted hereafter as phase A), suggesting the growth of a new phase (denoted hereafter as phase A′), whereas no appearance of phase A′ was observed when the charge cut-off potential was limited at 4.6 V. Interestingly, by clamping the potential at 4.8 V, we found that phase A′ grows at the expense of the parent A phase, and eventually becomes a pure phase after 5 h constant-voltage (CV) hold (Fig. 1a). This is for the first time that the mysterious phase A′ has been obtained phase pure, to the best of our knowledge, enabling its detailed structural characterization.

The structure of the A′ phase was determined by combined transmission electron microscopy (TEM), synchrotron XRD, and neutron powder diffraction (NPD). The electron diffraction (ED) patterns (Supplementary Fig. 4) reflect that the vast majority of Li-rich NMC phases (4.6 V charged (the A phase), 4.8 V charged (the A′ phase), 4.8 V charged and 2.0 V discharged) crystallize in the O3 $R\bar{3}m$ structure. For all phases, the [0 1 0] ED patterns demonstrate reflection splitting along the $c^*$ direction, which is the typical signature of the mirror-twinned O3 $R\bar{3}m$ structure with the (0 0 1) twin plane (Supplementary Figs. 4 and 6). The nanosized twinned domains are evident in the low magnification high-angle annular dark-field scanning TEM (HAADF-STEM) images (Supplementary Fig. 4). Besides sharp reflections of the O3 structure, diffuse spots are present in the [0 1 0] ED pattern of the A′ phase at the positions characteristic of the O1 structure (marked with vertical arrowheads in Supplementary Fig. 4b). In contrast to pristine Li-rich NMC[28], the [$\bar{1}$ 1 0] ED pattern of the A′ phase shows only very faint diffuse intensity lines from the "honeycomb" Li-M ordering (marked with horizontal arrowheads in Supplementary Fig. 4b), indicating that this ordering is largely suppressed. These results justify the validity of the simple O3 $R\bar{3}m$ structure model used for further Rietveld refinement of the A′ structure.

In agreement with ED data, synchrotron XRD pattern of the phase A′ can be indexed in an O3 $R\bar{3}m$ structure with lattice parameters of $a = 2.84194(11)$ Å and $c = 13.9722(14)$ Å. As the distribution of five species with different scattering factors (Li, Ni, Mn, Co, and cation vacancy) among two crystallographic positions of the $R\bar{3}m$ structure cannot be distinguished uniquely, combined neutron/synchrotron Rietveld refinement was carried out with O and Li content fixed to $Li_{0.05}MO_{1.85}$, as deduced from inductively coupled plasma-optical emission spectrometry (ICP-OES) and online electrochemical mass spectrometry (OEMS) (Supplementary Table 2), and assuming that only one of the three transition-metal (M) cations may migrate. Indeed, NPD provides a unique contrast between Mn ($b_{Mn} = -3.73$ fm), Ni ($b_{Ni} = 10.3$ fm), and Co ($b_{Co} = 2.49$ fm). The total occupancy of each M cation was restricted to its respective content in the pristine chemical formula. Cation migration to octahedral Li vacant sites and interstitial

tetrahedral sites were considered, but the latter was discarded based on difference Fourier maps and Rietveld refinements (Supplementary Fig. 5). The best fit (Fig. 1b, c) was obtained by populating vacant Li 3a sites with 0.079(2) Mn, hence a chemical composition of $[Li_0Mn_{0.08}][Li_{0.05}Ni_{0.13}Mn_{0.46}Co_{0.13}]O_{1.85}$ (square brackets refer, respectively, to octahedral 3a and 3b sites of the $R\bar{3}m$ structure). However, one should note two possible interpretations of such a refinement. The first corresponds to the above-written chemical formula and involves the presence of oxygen vacancies and undercoordinated metals. The second is to rewrite the chemical formula as $[Li_0Mn_{0.08}][Li_{0.05}Ni_{0.14}Mn_{0.50}Co_{0.14}]O_2$. This corresponds to an O3 structure without oxygen vacancies but "densified," because M/O ratio has increased compared to that of pristine $Li_{1.2}Ni_{0.13}Mn_{0.54}Co_{0.13}O_2$ (0.43 for the A′ phase, 0.40 for the pristine phase). The latter is in full agreement with the fact that the unit cell volume of the A′ phase is smaller compared to that of the pristine phase (97.730(11) Å³ for the A′ phase, 100.545(2) Å³

for the pristine phase), because of the c-lattice parameter shortening (from 14.25381(15) Å for the pristine phase to 13.9722(14) Å for the A′ phase) (Supplementary Tables 1 and 3). Note that Mn migration to vacant octahedral sites in Li layers was already theoretically proposed and experimentally observed[31–34]. Lastly, it is worth mentioning that the c/a aspect ratio in the A′ phase is very close to (but different from) that of an ideal cubic lattice ($c/a = 4\sqrt{3}/\sqrt{2}$), which can explain why this phase may have been previously misinterpreted in the literature as a spinel cubic phase. Altogether, these results clearly indicate that "densification" is not a surface but rather a bulk effect.

To enforce this structural analysis, we performed atomic resolution HAADF-STEM imaging of the A′ phase. The [0 1 0] HAADF-STEM images (Supplementary Fig. 7a) reflect the prevailing O3-type stacking of M layers, albeit with thin domains of O1-type stacking spanning over just few unit cells. More extended O1 domains are also observed (Supplementary Fig. 7b). In the O3 structure, the layered ordering of M cations is violated by M cation migration to vacant Li sites appearing as a nucleation of the $\{10\bar{2}\}$-oriented M cation layer at ~60° to the $\{0\,0\,1\}$-oriented $Li_{1-x}M_{2+x}$ layers. The O1 domains demonstrate more pronounced M cations occupation in octahedral interstices. The most remarkable picture of structural inhomogeneity in the A′ phase related to M cation migration is visualized with the $[\bar{1}\,1\,0]$ HAADF-STEM images (Fig. 2a). First inhomogeneity is related to the "honeycomb" Li-M ordering, which only pertains in part of the $Li_{1-x}M_{2+x}$ layers, whereas the layers with suppressed "honeycomb" ordering are frequently present (both variants are marked with arrowheads in Fig. 2a). Significant M cation migration occurs over whole >15 nm distance from the surface demonstrating a variety of local ordering, where the M cations occupy either octahedral Li sites or tetrahedral interstices (insets in Fig. 2a). The enlarged images of this local ordering, along with tentative cation distribution among tetrahedral and octahedral sites, are provided in Supplementary Fig. 8. These locally ordered sequences also manifest themselves in reciprocal space as very faint reflections between the diffuse intensity lines in the $[\bar{1}\,1\,0]$ ED pattern (Supplementary Fig. 4b), and can be considered as a prerequisite to the formation of spinel-type structure. Hence, we conclude that the A′ phase preferentially adopts the O3-type structure with M cations partially migrating to the empty octahedral or tetrahedral sites at the surface as well as in the interior parts of the crystallites causing bulk densification. This migration is associated with the A′ phase formation, and is substantially reversible as comparatively illustrated with the $[\bar{1}\,1\,0]$ HAADF-STEM images (Fig. 2b–d).

**$O_2$ release on deep oxidation.** To provide insights into the origin of A → A′ phase transition, we carry out OEMS measurements under various oxidizing conditions. The onset of $O_2$ release is observed at ca. 4.6 V (Fig. 3b and Supplementary Fig. 9) where the phase transition starts (Supplementary Fig. 3). Further pushing the oxidation by raising the potential to 4.8 V and applying either a CV hold or an OCV step (Supplementary Fig. 9) results in additional $O_2$ release. The cumulated $O_2$ release amounts, respectively, to 751.0 and 383.4 μmol g⁻¹ for cells charged with additional CV and OCV steps, as compared to 269.5 μmol g⁻¹ for the cell directly switched to discharge after reaching 4.8 V (Supplementary Fig. 9 and Supplementary Table 2). Overall, OEMS and operando XRD results show that the growth of phase A′ solely occurs after the onset of $O_2$ release (Supplementary Figs. 9 and 3), suggesting that $O_2$ release and A → A′ phase transition are strongly correlated. We further tracked down both the structural evolution and $O_2$ release upon subsequent cycling. Operando XRD study shows that the phase

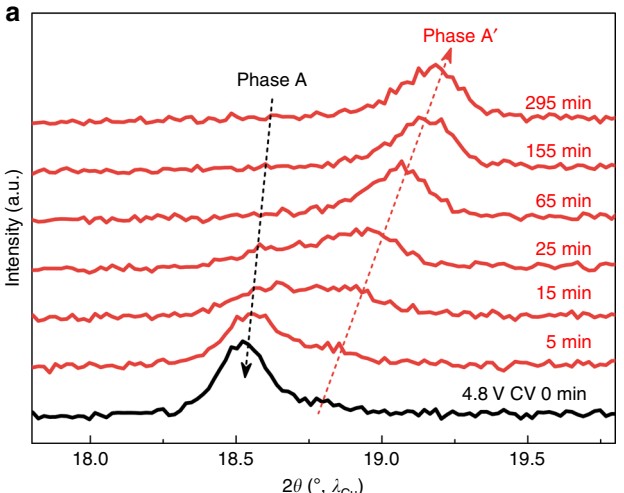

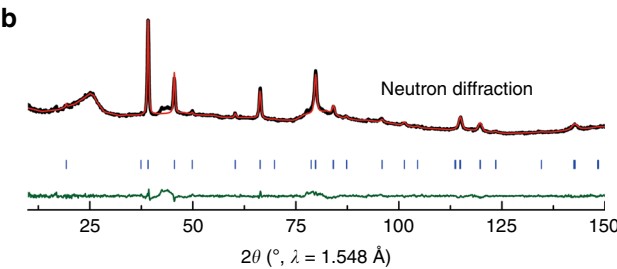

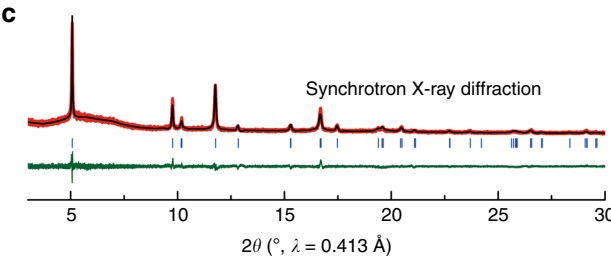

**Fig. 1 Lattice densification of Li-rich NMC on deep oxidation. a** Selective region of the operando XRD patterns (around the (0 0 3) diffraction peak) collected at the end of 4.8 V charge (denoted as 4.8 V CV 0 min) and during the 5 h CV hold at 4.8 V. Note that phase A refers to the charged Li-rich NMC phase (the parent (0 0 3) peak), whereas phase A′ refers to the new peak developed at the high-angle side of the parent (0 0 3) peak. Rietveld refinement of neutron powder (**b**) and synchrotron X-ray (**c**) diffraction for the A′ phase.

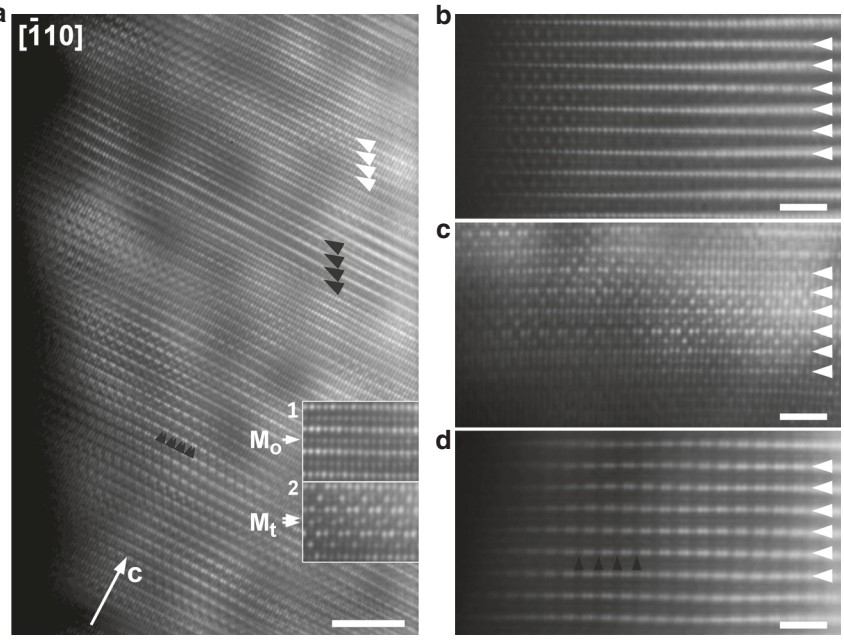

**Fig. 2 Local structure of Li-rich NMCs viewed with HAADF-STEM images. a** Overview [$\bar{1}10$] HAADF-STEM image showing structural inhomogeneity of the A′ phase. Scale bar indicates 1 nm. The $Li_{1-x}M_{2+x}$ layers are visible as rows of brighter dots perpendicular to the c-axis. "Honeycomb" Li-M ordering is visualized by pairs of bright dots with ~0.14 nm interdot separation (such as marked with pairs of small black arrowheads). The $Li_{1-x}M_{2+x}$ layers with pertaining "honeycomb" ordering (such as marked with large white arrowheads) alternate with the layers where this ordering is largely suppressed (such as marked with large black arrowheads). The interlayer spaces demonstrate pronounced HAADF intensity indicating migration of the M cations to either octahedral sites $M_o$ (see inset 1) or to the tetrahedral interstices $M_t$ (see inset 2, see Supplementary Fig. 8 for atomic position assignment). This migration occurs at the near-surface region as well (at >15 nm away from the surface). Near-surface [$\bar{1}10$] HAADF-STEM images of Li-rich NMC charged to 4.6 V (**b**, the A phase), 4.8 V (**c**, the A′ phase) and discharged to 2.0 V (**d**), scale bars indicate 1 nm. The M cation migration sets in at very thin 1–2 nm surface layer in the A phase, then it becomes very pronounced in the A′ phase and finally gets almost fully suppressed after 2.0 V discharge. The "honeycomb" Li-M ordering is largely restored in the discharged phase as demonstrated by the pattern of the bright dot pairs, but some disorder still remains indicated by residual intensity between the dot pairs (marked with black arrowheads in **c**).

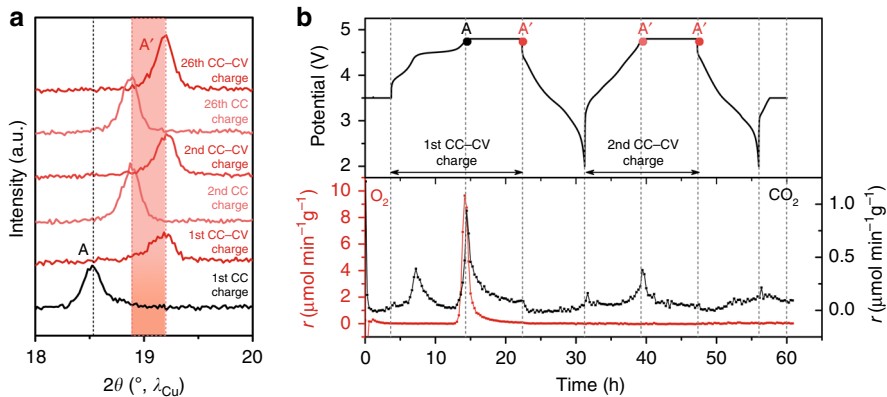

**Fig. 3 Operando XRD and OEMS studies of Li-rich NMC during the first cycle and subsequent cycling. a** Laboratory XRD patterns of Li-rich NMC collected at various state of charge in an operando XRD cell, which was cycled between 4.8 and 2.0 V, with an intermediate CV step at 4.8 V for 5 h at each cycle. **b** OEMS gas analysis during the first and second cycles of Li-rich NMC. The cell has been cycled at C/10 between 4.8 and 2.0 V, with a CV step at 4.8 V for 8 h. $O_2$ ($m/z = 32$) and $CO_2$ ($m/z = 44$) ion currents were recorded and then converted to gas evolution rates. The potential versus time curves is presented in the upper panel, and the evolved rates of $O_2$ (red line) and $CO_2$ (black line) in the units of $\mu mol\ min^{-1}\ g^{-1}$ are shown in the bottom panel.

A′ repeatedly forms at the end of each CC-CV charge until 26th cycle (the maximum number of cycle we have tried, Fig. 3a), whereas no $O_2$ release was observed in the second cycle (Fig. 3b). Altogether, these results imply that during the initial charge, the formation of phase A′, whose existence ranges from $x = 0.15$ to 0.05 (dashed square in Fig. 3a), is caused by the conjoint removal of $Li^+$ and O, which creates both cationic and anionic vacancies followed by lattice densification. Once formed, the new phase A′

can reversibly uptake and release $Li^+$ without having more anionic vacancies, hence $O_2$ release.

**Mn migration associated with the activation of oxygen redox on charge.** Bearing in mind that the A′ phase enlists Mn migration, we then investigated how the Mn migration progresses during charge by preparing several $Li_{1.2}Ni_{0.13}Mn_{0.54}Co_{0.13}O_2$/Li cells that

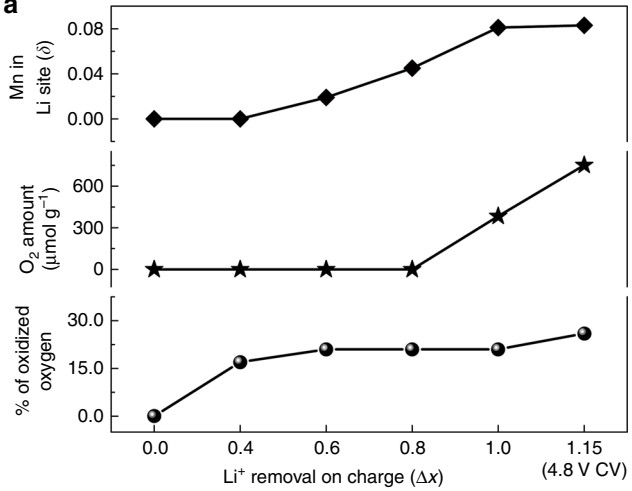

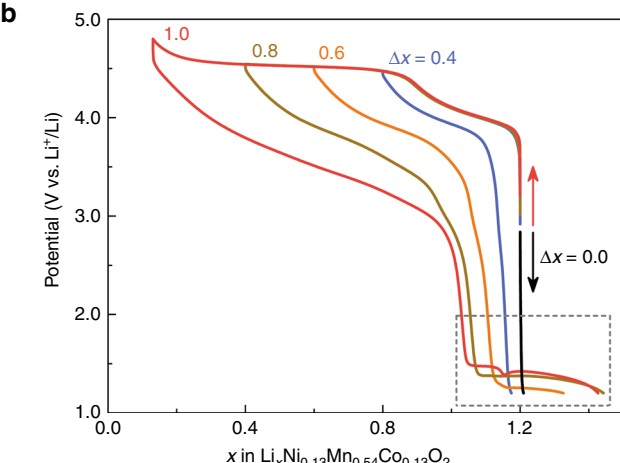

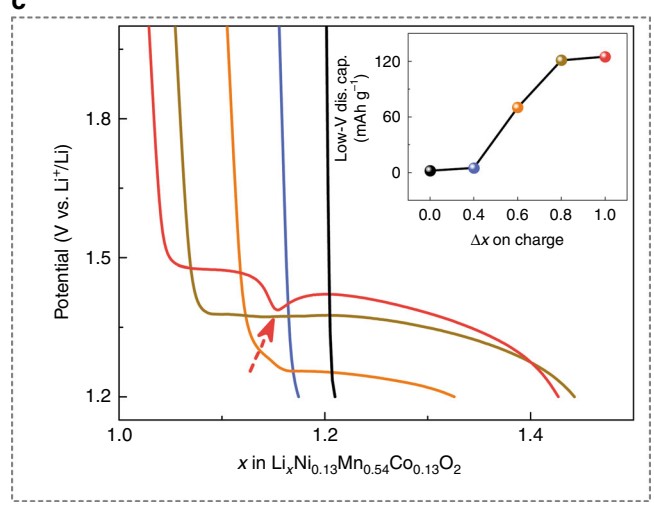

**Fig. 4 Mn migration associated with the triggering of anionic redox on charge and its consequence of low-potential electrochemical activities. a** The upper panel presents Mn migration in interlayer octahedral Li sites (δ) deduced from Rietveld refinement of synchrotron XRD patterns collected for various charged samples with the amount of Li$^+$ removal ($\Delta x$) ranging from 0.0 (pristine), 0.4, 0.6, 0.8, 1.0 up to 1.15 (with CV hold at 4.8 V for 8 h, denoted as "4.8 V CV"). The amount of released O$_2$ quantified by OEMS and the fraction of oxidized oxygen (defined as O$^{n-}$/(O$^{n-}$ + O$^{2-}$)) quantified from XPS are, respectively, plotted as a function of $\Delta x$ in the middle and bottom panels. **b** Potential-composition profiles of Li-rich NMC obtained by varying $\Delta x$ on charge from 0.0 (pristine, black line), 0.4 (purple line), 0.6 (orange line), 0.8 (gold line), to 1.0 (red line), while fixing the discharge cut-off potential at 1.2 V. The dashed gray square indicates the gradual appearance of low-potential electrochemical activities (2.0–1.2 V) upon pushing the extent of charge to trigger the anionic redox. **c** A magnified view of the potential-composition profiles in the low-potential region (marked with dashed gray square in **b**). The dash red arrow indicates the potential overshooting in discharge after charging up to $\Delta x =$ 1.0. The inset shows the amount of low-potential discharge capacities as a function of $\Delta x$. Note that the colors in **c** are quoted the same as in **b**.

the $\Delta x = 1.0$ charged sample at 4.8 V for 8 h to obtain the pure phase A′ did not change the amount of Mn migration within the accuracy of the method, but decreased the $c$-lattice parameter from 14.2058(4) to 13.9722(14) Å. This is consistent with the decrease in Li content from 0.02 (for $\Delta x = 1.0$ charged sample) to 0.005 (for 4.8 V CV charged sample), according to ICP-OES. X-ray photo-emission spectroscopy (XPS) measurements were carried out for the various charged samples to determine the fraction of oxygen involved in the anionic redox process. This fraction increases with $\Delta x$ alike the degree of Mn migration; hence, implying their correlation (Fig. 4a). In contrast, neither Mn migration nor the percentage of oxidized oxygen correlates with O$_2$ release (Fig. 4a).

**Extra low-potential electrochemical activity.** Besides exploring the harsh oxidizing conditions, we next investigated the effect of harsh reducing conditions on the electrochemical signatures of Li-rich NMC. At first, we fixed the discharge cut-off potential to 2.0 V and monitored the effect of state of charge on the subsequent discharge profile by opening the charge potential window stepwise (Supplementary Fig. 10). We observed that the S-type discharge profile becomes more pronounced as the charge potential reaches the high-potential plateau. Meanwhile, the first-cycle irreversible capacity increases with deeper oxidation. Strikingly, extra redox activity located within 1.5–1.2 V, whose capacity increases with $\Delta x$ on charge (Fig. 4b, c), gradually appears when the cells were further discharged to 1.2 V. This extra electrochemical activity is absent when the cell was directly started towards reduction ($\Delta x = 0.0$) or when the charge was limited to the cationic redox domain ($\Delta x = 0.4$), suggesting a straightforward correlation between the extra low-potential capacities and the extent of anionic redox process. A possible explanation could be rooted in Mn migration, which is observed in the charged phases provided the participation of anionic redox. We therefore revisit the Rietveld refinements of synchrotron XRD patterns collected for the various charged samples (Fig. 4a and Supplementary Fig. 11). Indeed, Mn migration appears only when charging to the anionic redox region, and progressively builds up with increasing state of charge. Moreover, no Mn migration is observed when the charge is limited to the cationic redox ($\Delta x = 0.4$), akin to the pristine phase ($\Delta x = 0.0$) (Fig. 4a). Altogether, our results suggest a robust correlation between the appearance of low-potential redox activity and the Mn migration associated with anionic redox participation on charge. Additionally, we

were electrochemically oxidized till $\Delta x = 0.4$, 0.6, 0.8, and 1.0, prior to sampling the electrodes after 120 h OCV for ex situ XRD ($\Delta x$ refers to the amount of Li$^+$ removal on charge or the amount of Li$^+$ insertion on discharge). By employing the same Rietveld methodology detailed before, we could deduce that while the $\Delta x = 0.4$ charged sample shows no Mn migration in interlayer octahedral sites (defined as δ in [Li$_{1.0-\Delta x}$Mn$_\delta$][Li$_{0.2}$Ni$_{0.13}$Mn$_{0.54-\delta}$Co$_{0.13}$]O$_2$), a progressive increase of Mn migration was observed with increasing $\Delta x$ from 0.6 up to 1.0 (Fig. 4a). Interestingly, further maintaining

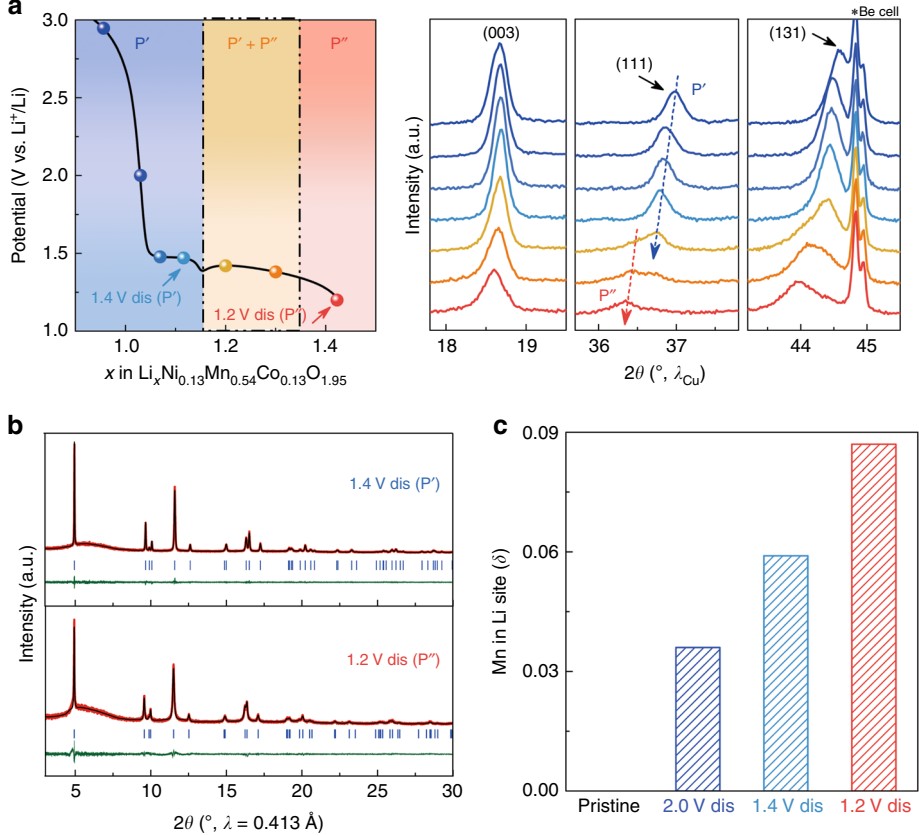

**Fig. 5 Structural evolutions during low-potential discharge process. a** Selective galvanostaic discharge curve (left side) and selective regions ((0 0 3), (1 1 1), and (1 3 1) diffraction peaks) of operando XRD patterns (right side) collected upon discharging Li-rich NMC from 3.0 down to 1.2 V, after 4.8 V charge and 3.0 V discharge (see Results for the complete discharge in Supplementary Fig. 13). Note that the dashed blue arrow marks the evolution of phase P′ and the dashed red arrow indicates the growth of phase P″, respectively. **b** Rietveld refinements of synchrotron XRD patterns for the samples collected after 1.4 V (phase P′) and 1.2 V (phase P″) discharge. **c** Evolution of Mn migration in interlayer octahedral Li sites (δ) in the pristine phase, and the phases collected at 2.0, 1.4, and 1.2 V discharge state after 4.8 V charge.

found that Mn migration can only be partially restored during discharge, since Mn migration (δ) of 0.0349(4) still remains for a 4.8 V charged and 2.0 V discharged sample (Supplementary Fig. 12). ED patterns and HAADF-STEM analysis of this phase suggest that the microstructure of Li-rich NMC is largely restored (Supplementary Fig. 4c and Fig. 2d). Hence, such a discharged phase, which slightly deviates from the pristine one, is denoted as P′ hereafter.

**Formation of a new phase P″ on deep discharge.** Further scrutinizing the low-potential redox region (below 2.0 V) enlists two domains separated by a potential overshooting (marked by a dash red arrow in Fig. 4c), which is usually reminiscent of the nucleation-growth process of a new phase. To study the structural changes in low-potential region, a $Li_{1.2}Ni_{0.13}Mn_{0.54}Co_{0.13}O_2$/Li operando XRD cell was charged to 4.8 V and discharged to 3.0 V, prior to being pushed down to 1.2 V while collecting XRD patterns (Fig. 5). A continuous initial shift of the P′ phase reflections (the (0 0 3) reflection shifts toward higher angles, whereas the (1 1 1) and (1 3 1) reflections shift toward lower angles) with gradual decrease in intensity till the potential overshooting (~1.4 V), indicating a solid solution insertion process. Upon further increase of $x$ (Li content), some reflections disappear at the expense of new ones that remain broad until becoming a single reflection, indicative of a new phase (denoted hereafter as P″) formation when the potential reaches 1.2 V.

Owing to the width of these reflections, we collected high-resolution synchrotron XRD patterns for 1.4 V (phase P′) and 1.2 V (phase P″) discharged samples (Fig. 5b). Rietveld refinements indicate that both phases remain in the $R\bar{3}m$ space group with lattice parameters smaller for the P′ phase ($a = 2.87386(2)$ Å, $c = 14.3442(2)$ Å, $V = 102.598(2)$ Å$^3$), larger for the P″ phase ($a = 2.90026(5)$ Å, $c = 14.3783(5)$ Å, $V = 104.739(4)$ Å$^3$), and the amount of Mn migration (δ) determined, respectively, to be 0.057 (2) and 0.085(2) (Fig. 5c). This increase in lattice parameters from P′ to P″ phase is consistent with the uptake of $Li^+$ upon discharging from 1.4 to 1.2 V ($Li_{1.14}Ni_{0.13}Mn_{0.54}Co_{0.13}O_{1.95}$ for the phase P′ and $Li_{1.44}Ni_{0.13}Mn_{0.54}Co_{0.13}O_{1.95}$ for the phase P″, as deduced by ICP-OES for Li content and OEMS for O content).

Low scattering power of Li prevents a fully reliable determination of its position by synchrotron XRD. However, previous studies have shown that $LiNiO_2$ upon reduction can uptake one more $Li^+$ leading to the $Li_2NiO_2$ structure, having all the $Li^+$ in tetrahedral sites[35,36]. Therefore, we can postulate, bearing in mind that all the octahedral sites are fully occupied in $Li_{1.2}Ni_{0.13}Mn_{0.54}Co_{0.13}O_2$, that the P″ phase most likely contains some $Li^+$ in tetrahedral sites, the only sites available to host $Li^+$ in excess of 1.2. Nevertheless, $Li_2NiO_2$ has a 1T structure (space group $P\bar{3}m1$) with ABAB oxygen stacking sequence, whereas P″ phase shares the same O3 structure, that is, the same ABCABC oxygen stacking sequence, as the pristine $Li_{1.2}Ni_{0.13}Mn_{0.54}Co_{0.13}O_2$. Therefore, another puzzle concerns why the over-lithiation of P′ to P″ phase does not induce an O3 to

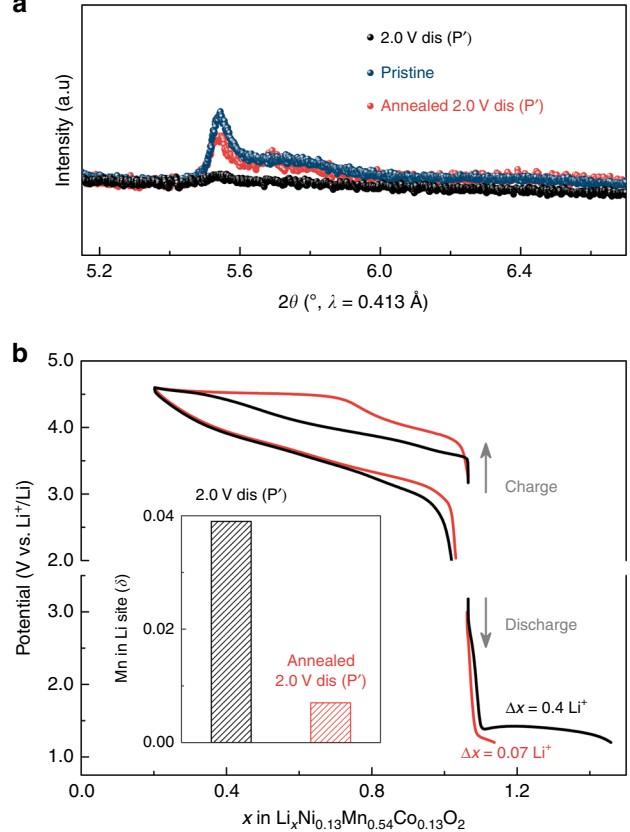

**Fig. 6 Reduced Mn migration (δ) and low-potential discharge capacity via mild-temperature annealing. a** Comparison of the superstructure peak in the pristine (marine blue symbol), 2.0 V discharged phase P′ before (denoted as "2.0 V dis (P′)," black symbol), and after (denoted as "annealed 2.0 V dis (P′)," red symbol) annealing. **b** Potential-composition profiles of the 2.0 V discharged phase P′ before (black lines) and after (red lines) annealing. Galvanostatic tests started on charge are presented in the upper panel, revealing that the charge profile was restored to its original staircase shape and its potential was recovered by mild-temperature annealing. Cells started on discharge shown that the amount of Li⁺ inserted (Δx) during the low-potential discharge process was drastically decreased by the heat treatment (~0.4 Li⁺ and ~0.07 Li⁺ for the 2.0 V discharged phase P′ before and after annealing, respectively). The inset shows that Mn migration (δ) has been drastically reduced to nearly zero after annealing, as deduced from the Rietveld refinements of synchrotron XRD patterns collected for 2.0 V dis (P′) phase and annealed 2.0 V dis (P′) phase (see the Rietveld refinements in Supplementary Fig. 14).

**1T phase transition.** Since a considerable amount of Mn (0.057 (2)) migrates to interlayer octahedral sites in the phase P′, we believe that these migrated Mn ions serve as "pillars" to stabilize the structure, preventing the rearrangement of the oxygen sublattice. This argument is in good agreement with previous reports, where $Li_{1.2}Cr_{0.4}Mn_{0.4}O_2$ phase with Cr migration in interlayer octahedral sites could retain its original *ccp* oxygen stacking during overlithiation[37,38].

**Role of Mn migration on the low-potential electrochemical activity.** To further confirm our proposed Mn migration-driven mechanism, we performed, inspired by the work of Singer et al.[23], a heat treatment of the P′ phase (prepared by charging to 4.6 V and discharging to 2.0 V) under Ar atmosphere at 250 °C for 1 h. Synchrotron XRD of the annealed sample shows that such mild-temperature annealing recovers the superstructure diffraction

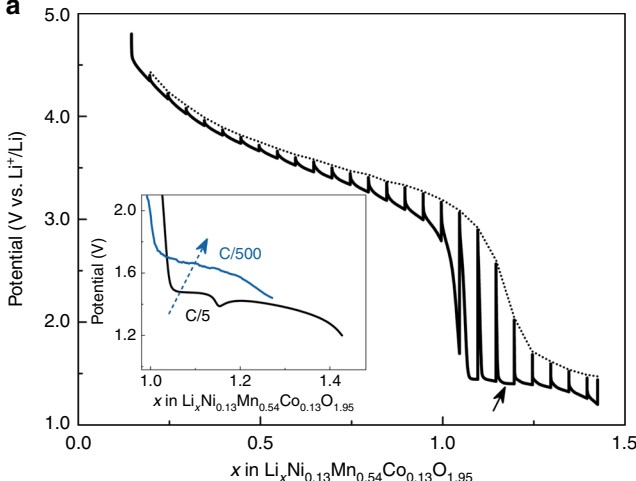

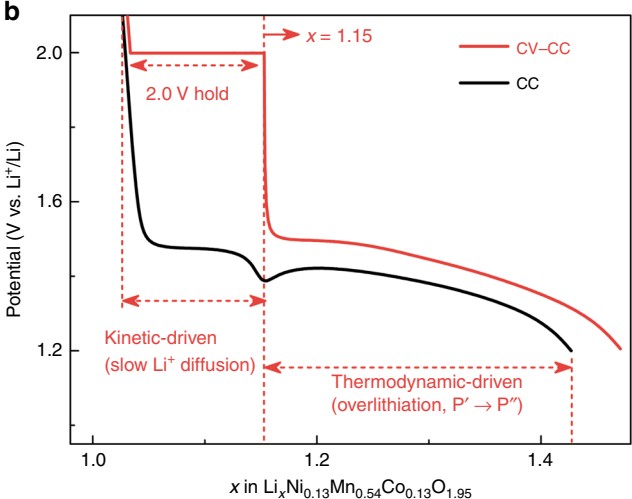

**Fig. 7 Thermodynamic and kinetic aspects of the low-potential discharge process. a** Galvanostatic intermittent titration technique (GITT) of Li-rich NMC during the first discharge to 1.2 V, after 4.8 V charge. The relaxation process was controlled by either dV/dt ≤ 0.01 mV s⁻¹ or time = 4 h. The black arrow marks the potential overshooting. The inset shows the cells discharged at current rates of C/5 (black line) and C/500 (marine blue line). **b** Potential-composition profiles of Li-rich NMC at low-potential region: two identical cells were first charged to 4.8 V, prior to being discharged with either a constant current (denoted as "CC," black line) or a constant current-constant voltage (denoted as "CC-CV," red line) protocol with the CV hold applied at 2.0 V over 40 h till the current decays to 6 μA.

peaks from the "honeycomb" Li-M ordering within $Li_{1-x}M_{2+x}$ layer (Fig. 6a). More importantly, Rietveld refinements indicate that Mn migration (δ) is reduced after annealing from 0.038(2) to nearly zero (0.007(2)) (inset in Fig. 6b). Meanwhile, electrochemical tests reveal that the annealed sample shows a staircase charge profile pertaining to the pristine phase, and a negligible extra capacity of ~0.07 Li⁺ as opposed to ~0.4 Li⁺ for the non-annealed phase (Fig. 6b). Altogether, the amount of Mn migration and the low-potential discharge capacity decrease in a similar manner after annealing, unambiguously confirming that the extra lithiation at low potentials is related to Mn migration.

**Thermodynamic and kinetic aspects of the low-potential discharge process.** Now it pertains to gain better insights into the origin of the large potential drop (occurring at ca. x = 1.0) needed

**Fig. 8 Schematic of the A → A′ → P′ phase transition.** Delithiation and oxygen evolution create vacancies in both cationic and anionic sublattices upon charge to above 4.6 V in the first electrochemical cycle. The oxygen vacancies are refilled implying their long-range migration from bulk to surface with subsequent annihilation. The vacant Li sites are partially refilled with the M cations causing their appearance in the octahedral interlayer sites and suppressing the "honeycomb" Li-M cation ordering. As a result, M/O atomic ratio increases that is considered as "densification." During subsequent discharge, the M cations partially move back to their original positions and the vacant cationic sites are filled with Li$^+$. Li: yellow symbol; Li vacancy: dash yellow square; transition metals (Ni, Mn, Co): blue symbol; transition-metal vacancies: dash blue square; oxygen: red symbol; oxygen vacancy: red dash square.

to insert additional Li$^+$. Possible electronic limitations can be excluded since increasing the electrode carbon content from 20 to 40 wt% did not change the amplitude of potential drop (Supplementary Fig. 15). Galvanostatic intermittent titration technique (GITT) was additionally performed to study the thermodynamic–kinetic aspects of the low-potential discharge process (Fig. 7a). The large polarization, observed before the o-vershooting (marked by black arrow in Fig. 7a), suggests that the potential drop in this regime is nested in kinetically hindered Li$^+$ diffusion. This result is consistent with the observation that the amplitude of potential drop can be reduced by either decreasing the discharge current rate (C/500 instead of C/5, inset in Fig. 7a) or raising the operation temperature (from room temperature to 55 °C and 75 °C, Supplementary Fig. 16). We further exploit this kinetic limitation by applying a CV hold at 2.0 V over 40 h until the current decays to nearly zero (~6 μA) and succeed in inserting ~0.12 extra Li$^+$ (Fig. 7b), indicating that a major part (ca. 70%) of the first-cycle irreversible lost Li$^+$ (~0.17) observed upon constant current (CC) cycling is due to kinetic inhibition and can be recovered by performing CC-CV-type cycling. Beyond the overshooting, a thermodynamically driven P′ → P″ phase transition with Li$^+$ insertion being charge compensated by Co and Mn redox, as deduced from ex situ X-ray absorption spectroscopy (XAS) (Supplementary Fig. 17).

**Practicability of cycling Li-rich NMC under harsh oxidizing and reducing conditions**. We finally evaluated the practical benefits, if any, of cycling Li-rich NMC under harsh oxidizing (Supplementary Fig. 18) or reducing (Supplementary Fig. 20) conditions. For the former case, three identical Li$_{1.2}$Ni$_{0.13}$Mn$_{0.54}$-Co$_{0.13}$O$_2$/Li cells were cycled at C/3 between 4.8 and 2.0 V, but with different first-charge protocols: one cell was directly switched to discharge with phase A forming at the end of the first charge, whereas either an intermediate OCV step or a CV hold was imposed to the other two cells to trigger the formation of phase A′. Phase A′ exhibits larger first-cycle irreversible capacity and an initial voltage drop as compared to phase A (Supplementary Fig. 18a), while upon cycling, the persistence of voltage decay and capacity fading (Supplementary Figs. 18b, c). From an electro-chemical perspective, the initial voltage drop can be explained by the reduction of Co and Ni during OCV or CV step, as revealed by in situ XAS (Supplementary Fig. 19). The observed capacity drop after imposing OCV or CV step is closely associated with the contraction of M layers and the growing Mn migration upon gradual A → A′ phase transition, which causes kinetic hindrance in Li$^+$ intercalation. Moreover, the formation of phase A′ is

accompanied with O$_2$ release, which is known to induce particle cracking that could also contribute to the performance dete-rioration. Another three cells were assembled and cycled at C/5 by fixing the charge cut-off potential to 4.8 V, but varying their dis-charge cut-off potential (2.0, 1.4, or 1.2 V, Supplementary Fig. 20). Around 1.3 Li$^+$ (403 mAh g$^{-1}$) can be reversibly uptaken and released on further discharge to 1.2 V, as compared to 1.0 Li$^+$ (322 mAh g$^{-1}$) and 0.86 Li$^+$ (272 mAh g$^{-1}$) during 1.4 and 2.0 V discharge, respectively. Although most of the extra capacity is delivered at low potentials, it translates into a 20% of increase in energy density. Nevertheless, note that discharging to 1.2 V leads to faster capacity decay, and therefore to a comparable energy density to that of 2.0 V discharge after 25 cycles (Supplementary Fig. 20b, d), which is not attractive application-wise.

## Discussion

We enlarge the cycling window for Li$_{1.2}$Ni$_{0.13}$Mn$_{0.54}$Co$_{0.13}$O$_2$ electrode to both high and low potentials, and unravel novel structural evolution mechanisms that are discussed next within the context of numerous literatures. We identify a not-yet-reported structurally densified pure phase A′, which forms under harsh oxidizing conditions within the bulk of the material not only at the surface, in contrast to previous beliefs. Numerous reports evidenced the hint of a new phase, described as a densified surface layer formed at the crystal surface after oxygen loss towards the end of charge[7,39]. However, doubts have been expressed about this new structure[24,40], hence the ongoing studies on the topic stressing the existence of Li-driven dislocations, defects, transition-metal migration, and so on[8,23,41,42]. Although elegant, such findings fail to capture the complete mechanism as the fully charged pure-phase material remains inaccessible. This limitation was here amended by applying a CV charge step at 4.8 V. By combing laboratory and synchrotron XRD, NPD, TEM, OEMS, and ICP-OES, we can propose the following sequence of events for the first electrochemical cycle of Li$_{1.2}$Ni$_{0.13}$Mn$_{0.54}$-Co$_{0.13}$O$_2$ (Fig. 8).

First, there comes the oxidation of Ni$^{2+}$ and Co$^{3+}$ till Δx = 0.4, followed by the oxidation of anions that is accompanied with Mn migration until 4.6 V (Δx < 1.0). Afterwards, while some Li$^+$ can still be removed to reach nearly the fully delithiated phase, O$_2$ evolution turns on along with cationic vacancies filled by migrated M cations, eventually leading to a densified bulk phase with the presence of disorder rocksalt-spinel surface layer as frequently reported. Densification implies that oxygen vacancies formed during O$_2$ evolution migrate from bulk to surface with their subsequent annihilation causing increase in M/O atomic

ratio. From thermodynamic perspective, densification associated to increase of M/O ratio is always favorable[43]; thus, its appearance after either OCV or CV step is indicative of a kinetically limited growth of the thermodynamically stable densified phase A′, most likely due to sluggish migration of oxygen vacancies. Although there are indications that energy barriers for O vacancy migration in Li-rich layered oxides are lower than those in the stoichiometric $LiMO_2$ phases[44], these barriers can hardly be precisely computed because of too many unknowns (the exact nature of the migrating species ($O^-$, $O^{2-}$, $O_2^{n-}$, etc), possible clustering of O and cation vacancies, and multiple possible diffusion pathways). A pure phase A′ can only be obtained when sufficient amount of $Li^+$ is removed and when sufficient time is given, which might explain why the phase A′ was never reported before in a pure form. Upon discharge, the anionic and cationic networks of the densified phase are conjointly reduced with only part of the migrated cations returning to their initial positions, explaining the difference between the discharge phase P′ from the pristine phase even with additional CV discharge step. Upon recharge, these abovementioned scenarios repeat themselves by reversibly cycling the A′ phase (in this study up to 26 cycles), with one exception, that is, $O_2$ evolution occurs only in the first cycle (Fig. 3).

By applying a CV discharge step, we recover part of the first-cycle irreversible capacity loss. This feature is not specific to Li-rich NMC; similar conclusions were reached by Kasnatscheew et al.[45] and Zhou et al.[46] when studying Li-stoichiometric materials ($LiNi_{1/3}Mn_{1/3}Co_{1/3}O_2$, $LiNi_{0.8}Mn_{0.1}Co_{0.1}O_2$). Indeed, decrease in $Li^+$ diffusion coefficient at high state of discharge was commonly observed in many layered oxides[46,47], and is likely due to the contraction of $c$-lattice parameter as well as the inhomogeneous lithiation leading to high Li concentration (not enough vacancy) at the surface of the particles[48]. The feasibility of injecting extra $Li^+$ at low potentials (below 2.0 V) is not specific to Li-rich NMC either, as such a possibility has been reported for other Li-rich[37,38] and Li-stoichiometric[35,49] layered oxides. The overlithiation of these layered oxides generally results in a $Li_2MO_2$-type phase, as confirmed by Robert and Novák[35]. In contrast, what is unique about the $Li_{1.2}Ni_{0.13}Mn_{0.54}Co_{0.13}O_2$ phase is that (i) additional $Li^+$ cannot be injected into the pristine material by directly lowering the potential till 1.2 V, (ii) the low-potential electrochemical activity can only be activated if the compound has been charged sufficiently to trigger oxygen redox, and (iii) the overlithiation does not lead to an O3 to 1T phase transition. At first, it could be argued that this specific behavior is associated with the presence of $Li^+$ within the $MO_2$ layers in Li-rich NMC that modifies the respective stability of P′ versus P″ phase. However, this does not hold as other pristine Li-rich layered oxides already demonstrated the feasibility to reversibly uptake extra $Li^+$ ($Li_{2+x}IrO_3$, $Li_{3+x}IrO_4$)[18,19], through $R\bar{3}m$ to $P\bar{3}m1$ structural transition that doubles the number of available sites in Li layers. A more reasonable explanation could be rooted in the observed Mn migration triggered by the anionic redox activity, which unbalances the electrostatic interactions between the $MO_2$ layers together with Li intra-interlayer repartition. Moreover, the presence of a few Mn ions at interlayer Li sites could act as "pillars" preventing the gliding of layered planes, which is needed to obtain a 1T structure as previously observed for $Li_2NiO_2$[36].

Application-wise, exploiting the A → A′ phase transformation does not improve neither capacity decay nor voltage fade. Although we showed the positive effect of applying a CV step at 2.0 V for recovering part of the initial irreversible capacity loss. However, neither reaching the full formation of the P′ phase nor the exploitation of the low-potential activity provides substantial

improvements regarding voltage fade and hysteresis pertaining to Li-rich NMC. Moreover, none of the extreme cycling conditions used in this study were sufficient to retrieve Mn migration, with the exception of an annealing procedure at 250 °C, which was applied to the discharged sample. An obvious direction to better understand the role of cation migration on the electrochemical performance of Li-rich NMC could reside in the assembly of solid-state batteries that can be operated at various temperatures.

In summary, the results demonstrate the fundamental benefit of operating cells under harsh electrochemical conditions to deepen our understanding about Li-rich NMC in terms of $O_2$ release, phase evolution, cation migration, and oxygen redox activity with the establishment of a few key correlations that could guide the design of Li-rich phases.

## Methods

**Material synthesis and characterization.** Li-rich NMC powders were synthesized by a two-step process involving the co-precipitation of carbonate precursor followed by two annealing steps[28]. Synchrotron XRD measurements were performed on the 11-BM beamline of the Advanced Photon Source at Argonne National Laboratory, with a wavelength of 0.4128 Å. NPD pattern was collected with the high-resolution powder diffractometer SPODI in Forschungs-Neutronenquelle Heinz Maier-Leibnitz (FRM II) at Technical University of Munich with a wavelength of 1.5482 Å. Operando XRD measurements were carried out using an air-tight electrochemical cell, equipped with a Beryllium window in a BRUKER D8 Advance diffractometer with Cu Kα radiation source ($\lambda_{K\alpha1} = 1.54056$ Å, $\lambda_{K\alpha2} = 1.54439$ Å). ED patterns, HAADF-STEM images, annular bright-field scanning TEM images were obtained with an aberration-corrected Titan G3 TEM operated at 200 kV. TEM samples were prepared in an Ar-filled glovebox by crushing the crystals in a mortar with dimethyl carbonate as a solvent and depositing drop of suspension onto holey-carbon-coated copper grids. A specialized Gatan vacuum transfer holder was used for transferring samples to the TEM column without contacting air.

**Electrochemical tests.** All electrochemical characterization was performed in CR2032-type coin cells, which were assembled in an Ar-filled glovebox with metallic Li as negative electrode and two glass fiber separators (Whatman GF/D, UK) soaked with LP30 electrolyte (1 M $LiPF_6$ dissolved in 1:1 v/v ethylene carbonate/dimethyl carbonate, Solvionic, France). A blend of Li-rich NMC and carbon Super P loose powder at a 4:1 mass ratio was hand ground for 15 min before being used as positive electrode. Galvanostatic cycling tests were performed at current rate of C/5, unless specified otherwise. All the ex situ samples were prepared using Swagelok-type cells with similar cell stacking described above. The as-recovered samples were washed with dimethyl carbonate at least three times before being dried under vacuum. The near-equilibrium potential profiles of the first discharge were obtained using galvanostatic intermittent titration technique (GITT, every $\Delta x = 0.05$) with the relaxation process being controlled by either $dV/dt \leq 0.01$ mV s$^{-1}$ or time = 4 h.

**OEMS measurements.** An in-house developed cell was used for OEMS experiments[50]. A slurry consisting of 72% of Li-rich NMC, 18% of carbon Super P, and 10% of polyvinylidene fluoride or polyvinylidene difluoride binder was coated on one side of a Celgard 2400 separator (22 mm in diameter) with a wet thickness of 200 µm. Cells were assembled with Li discs of 20 mm in diameter as the negative electrode and LP30 electrolyte of 120 µL in volume.

**X-ray photoelectron spectroscopy.** Loose powder consisting of 90% of Li-rich NMC and 10% of carbon Super P was cycled to the desired state of charge. XPS measurements were carried out with a Kratos Axis Ultra spectrometer using focused monochromatic Al Kα radiation source ($h\nu = 1486.6$ eV). The XPS spectrometer was directly connected through a transfer chamber to an Ar dry box to avoid moisture/air exposure of the samples. The analyzed area of the samples was $300 \times 700$ µm$^2$. Peaks were recorded with constant pass energy of 20 eV. For the Ag $3d_{5/2}$ line, the full width at half-maximum was 0.58 eV under the recording conditions.

**X-ray absorption spectroscopy.** Self-standing films consisting of 74% of Li-rich NMC, 18% of carbon Super P, and 8% of polytetrafluoroethylene (PTFE) were cycled to the desired state of charge and state of discharge. After washing with dimethyl carbonate and drying under vacuum, the cycled electrodes were then placed between two layers of Kapton tape and sealed under Ar atmosphere in an air-tight transparent plastic bag. Ex situ and in situ XAS measurements at the three transition-metal (Ni, Co, Mn) K-edges were performed in transmission mode at the ROCK beamline at synchrotron SOLEIL (France).

**Mild-temperature heat treatment of the cycled Li-rich NMC electrodes**.
Positive electrode comprising Li-rich NMC and carbon Super P loose powder at a 4:1 mass ratio was charged to 4.6 V prior to being discharged to 2.0 V in a Swagelok-type cell. The discharged sample was then recovered in an Ar-filled glovebox, washed three times with dimethyl carbonate, and dried under vacuum before being annealed under Ar atmosphere at 250 °C for 1 h. After heat treatment, the as-obtained powder electrode was re-assembled into a CR2032-type coin cell against metallic Li to evaluate the impact of heat treatment on the electrochemical properties.

## Data availability

The data that support the findings of this study are available from the corresponding author upon reasonable request.

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

## Acknowledgements

We thank Sathiya Mariyappan for preparing Li-rich NMC powder and for her fruitful discussions, Thomas Marchandier for his help on XAS experiment, and Biao Li for his help on XPS sample preparation, together with Ronghuan Zhang for helpful discussions. We are also grateful to Elodie Salager and Benjamin Porcheron for NMR experiments that we decided to not include. J.-M.T. acknowledges funding from the European Research Council (ERC) (FP/2014)/ERC, Grant Project 670116-ARPEMA). A.M.A. is grateful to Russian Science Foundation for financial support (Grant 20-43-01012). Access to the TEM facilities was granted by Advanced Imaging Core Facility (AICF) of Skoltech. W.Y. acknowledges a fellowship from the China Scholarship Council (CSC) to perform this work at Sorbonne Université and Collège de France. Use of the 11-BM mail service of the APS at Argonne National Laboratory was supported by the US Department of Energy under contract No. DE-AC02-06CH11357 and is gratefully acknowledged.

## Author contributions

W.Y., A.G., and J.-M.T designed the experiments; W.Y. carried out the electrochemical measurements, laboratory XRD experiments, and prepared all the samples; G.R. analyzed the crystal structures and diffraction patterns; A.M.A. carried out the TEM studies; A.S. performed NPD experiment; L.Z. and S.T. performed the OEMS measurements; A.I. analyzed the XAS spectra; D.F. collected and analyzed the XPS spectra; D.G. conducted the ICP-OES experiments; W.Y., G.R., A.M.A., and J.-M.T. wrote the manuscript with the contributions of all the authors.

## Competing interests

The authors declare no competing interests.
