## [Peer Review File · Nature Communications]

Reviewers' comments:

Reviewer #1 (Remarks to the Author):

This work reports a very detailed investigation on the structural change during the deep charge and deep discharge of lithium-rich cathode material. It was found that the migration of Mn, coupled with the loss of oxygen, is closely associated with the extra capacity offered by the lithium-rich cathode. Detailed structural characterization also shed light on the phase transformation pathway when deeply delithiated at 4.8V and overlithiated at 1.2 V.

In the discussion session, the authors also discussed and application potential of the extra capacity. It will be helpful if the authors also comment on the negative side of the extra wide potential swing for real applications.

Overall, this is a well prepared work with deep insight. It is recommended for publication with minor revision.

Reviewer #2 (Remarks to the Author):

This work reports the phase evolution in Li-rich layered oxide under harsh condition. The authors report carefully designed experiments and the most distinct finding is a densified phase in the material bulk. The formation of a new phase under extreme oxidation is clear. However, the claim of this new phase in the bulk, not simply on the surface as reported before, is questionable. Regardless, the concern is the significance of the discovery of this densified phase, which results in no benefit in cell performance at all. All the other claims in this work, regarding the strategy to mitigate the first cycle irreversible capacity and recover the structural ordering by annealing are known for Li-rich layered oxides. This manuscript is not suitable for the publication at Nat. Commun.

Below are the questions and comments on the technical aspects of this work:

Regarding the claim of densified phase in the bulk material, the experimental evidence is not convincing. This claim is mainly drawn based on Rietveld refinement and STEM. For Rietveld refinement, the authors present two scenarios according to the chemical formula determined by ICP and OEMS, one has oxygen vacancies and the other does not, instead, it represents a densified phase with increased M/O ratio. The authors argue the second scenario is the case based on lattice parameters. What is the determining factor to exclude the first scenario? Also, how can the author draw the conclusion it is a bulk phenomenon rather than surface based on Fig. 1f? Where is the STEM image taken in the bulk rather than showing a schematic bulk in Fig. 1f? What is the scale in these STEM images being considered?

A whole-paragraph discussion in Page 7 is based on Fig. S4 and Fig. S5. The quality of the STEM images presented is very poor. The contrast here almost makes it impossible to follow the discussion on TM occupancy. What is the particle size of the pristine material? The bulk information presented in Fig. S4c is at most 10 nm.

The authors discussed the effect of OCV at high voltage (4.8 V), it is important to compare the electrochemistry of this material at various testing protocols. From Fig. S16, OCV or CV at 4.8 V results in a significant capacity drop and some voltage decay. What is the origin of such key difference? What is the correlation with phase A and A'?

For comparison, what is the STEM of phase A at the end of 4.6 V charge? Do the authors anticipate the contrasting observation between phase A and phase A' (both bulk and surface) in comparison of 4.6 CC and 4.8 CC-CV?

The authors claimed phase A' becomes reversible after the first charge. In Fig. S10, does Mn migration only remain at the surface or in the bulk as well? If in the bulk, the Mn cation trapped in the interlayer octahedral sites in Figure S10 should be highlighted to guide the readers.

Reviewer #3 (Remarks to the Author):

The paper by Yin et al reports on a study of a prototypical Li-excess compound ($\text{Li}_{1.2}\text{Ni}_{0.13}\text{Co}_{0.13}\text{Mn}_{0.54}\text{O}_2$) at extreme states of charge and discharge. The study reveals the emergence of a new phase, referred to as A', when the compound is held at 4.8 volt for 5 hours. The authors refine the A' phase as being O3 with some Mn having migrated from the transition metal layer to the Li layer. Irreversible capacity loss when charged to high voltage is recovered when the compound is cycled and held at very low voltages, leading to the formation of what the authors refer to as P' and then P''. The discovery of the A' phase at high voltages and the peculiar capacity recovery at low voltages are to my knowledge new results and provide invaluable clues to enable a better understanding of the electrochemistry of Li-excess compounds.

The paper could be suitable for publication in Nature Communications after the authors address the following points:

1. Throughout the paper the A' phase is referred to as a "densified phase". While this may be true, no clear evidence is provided in the paper. The arguments used to conclude that the A' phase is "densified" are not convincing or logical (bottom of page 6). As I understand the argument, it is concluded that A' is densified since its unit cell volume is smaller than that of the pristine phase. Most layered intercalation compounds contract at high states of charge (certainly along their c-axis), and this occurs without any densification (i.e. transition metal cations filling vacant sites and oxygen vacancies diffusing to the surface). More convincing evidence would be based on a careful refinement of site occupancies. If the authors insist on referring to the A' phase as a "densified phase" throughout the paper (as is the case in the current version), they need to provide stronger evidence for this statement.

2. The twinning at high states of charge is interesting and likely of importance to better understanding structural changes that occur at high states of charge. However, the paper is not clear in its description of the twinned structure. For example, it refers to "the mirror-twinned rhombohedral structure with the (001) twin plane." However it is not clear whether the (001) indices are for a rhombohedral unit cell, or for the hexagonal unit cell of the O3 crystal. Can the authors consider adding a schematic that illustrates the crystallography of the twins and how they may relate to the original A phase?

3. In the refinement of the A' phase, was tetrahedral Mn occupancy considered? Very few details are provided about the refinements. Are the authors able to speculate on the oxidation state of the tetrahedral Mn in the surface layer? It is not clear why the authors rule out tetrahedral Mn in the bulk phase. It seems that similar features that are attributed to tetrahedral Mn at the surface are also apparent in the bulk (Figure 1f and S4).

4. The HAADF-STEM images (e.g. S4) appear to show void formation, similar to that observed by Yan

et al Nature Nanotechnology 602 (2019). Can the authors comment on this (i.e. confirm or rule out that voids have in fact emerged within the bulk of the A' phase)?

5. Can the authors comment on the recent study by Taylor et al JACS, 141, 7333 (2019), which also shows recovered capacity at very low voltage (~ 1.7 V). In this study though, there was clear evidence of the reduction of peroxo-molecules. Could something similar be occurring in Li-excess Mn containing compounds?

6. The authors measure O₂ release and provide quantitative values. Can the authors translate these numbers into an estimate of how much the crystals have densified?

Reviewer #1:

1. In the discussion session, the authors also discussed the application potential of the extra capacity. It will be helpful if the authors also comment on the negative side of the extra wide potential swing for real applications.

Response: Our findings regarding practical applications are twofold. On the positive side, we demonstrate the feasibility to recover part of the first cycle capacity by doing a Constant Voltage at 2 V during discharge. On the negative side, we show that operating the cells under harsh oxidizing or reducing conditions leads to some penalties in terms of voltage hysteresis (Figure S18a), faster capacity decay (Figure S18b) and lower average discharge voltage (Figure S18c), which diminish the energy output. Faster capacity decay is also observed upon pushing the low-voltage discharge (Figure S20b).

Reviewer #2:

The authors report carefully designed experiments and the most distinct finding is a densified phase in the material bulk. The formation of a new phase under extreme oxidation is clear. However, the claim of this new phase in the bulk, not simply on the surface as reported before, is questionable. Regardless, the concern is the significance of the discovery of this densified phase, which results in no benefit in cell performance at all. All the other claims in this work, regarding the

strategy to mitigate the first cycle irreversible capacity and recover the structural ordering by annealing are known for Li-rich layered oxides. This manuscript is not suitable for the publication at Nat. Commun.

Response: This general comment touches on different points that are addressed separately.

→ We can understand the referee's concerns about our claim of bulk densification in Li-rich NMC which was never reported before. This is the reason why we have done in-depth combined XRD and HRTEM analysis to back-up the validity of our statement. However, this was not enough for referee #2 who wants supplementary explanation, evidence and even additional data that we are providing in his/her detailed specific question below.

→ The referee is right by telling our new findings (isolation of a new A' phase and extra capacity at low voltage) are not beneficial for improving the electrochemical performance, as we clearly stated in the discussion. However, should I conclude that any fundamental advances should lead to practical benefits?

→ The strategy to mitigate the first cycle irreversible capacity by constant voltage is known for conventional layered oxides, but not for Li-rich layered oxides to our best knowledge.

→ Turning to the strategy to recover the structural ordering it is indeed known (Shirley Meng's work dealing with temperature-driven annealing defects) as mentioned by the referee and this was clearly stated in our original version.

However, the novelties of our work resides in providing evidences for i) the slow Li⁺ diffusion caused by Mn occupation in octahedral Li sites, instead of O_{2(gas)} release as the primary cause of the first-cycle irreversible capacity in Li-rich NMC material; ii) the importance of Mn occupancy in octahedral Li sites in governing the extra Li⁺ uptake at low potential and iii) unlike for other layered oxides, the overlithiation of Li-rich NMC does not result in O3 → 1T phase transition thanks to the presence of Mn between the layers which helps stabilizing the structure.

1. Regarding the claim of densified phase in the bulk material, the experimental evidence is not convincing. This claim is mainly drawn based on Rietveld refinement and STEM. For Rietveld refinement, the authors present two scenarios according to the chemical formula determined by ICP and OEMS, one has oxygen vacancies and the other does not, instead, it represents a densified phase with increased M/O ratio. The authors argue the second scenario is the case based on lattice parameters. What is the determining factor to exclude the first scenario? Also, how can the author draw the conclusion it is a bulk phenomenon rather than surface based on Fig. 1f? Where is the STEM image taken in the bulk rather than showing a schematic bulk in Fig. 1f? What is the scale in these STEM images being considered?

Response: The comment concern two points that we will address one after the other.

a) First, the question is whether the A' densified phase is really a bulk and not only surface effect. Based on our analysis of the A' phase from X-ray and neutron diffraction, we are confident that this is bulk. Indeed for X-ray, we used a small wavelength of 0.41 Å and the measurement is done in Debye-Scherrer geometry (transmission through a capillary), the powder pattern we obtained therefore comes from the bulk. For neutrons it is even more true since neutrons penetrate the whole sample, and the measurement was done on a large volume sample (400 mgs) which was especially prepared for neutrons using a large electrochemical cell.

b) Secondly, the referee asks what are our arguments to exclude an oxygen-vacancies containing

compound. We think that a densified phase is much more likely because 1) the volume of the A' phase is smaller than the A phase (based on diffraction analysis) 2) the formation of phase A' from phase A is rather difficult since one has to maintain at high voltage and wait for several hours, this is consistent with a rather slow diffusion process (oxygen vacancies diffuse in the structure to reach the surface; therefore the bulk is densified) 3) having a structure with a significant number of oxygen vacancies would mean that a certain number of transition metals (Ni, Mn, Co) are under-coordinated (less than 6 O neighbors). We have rewritten this part of the paper.

c) Third, we have re-drawn the HAADF-STEM image of the A' phase (now Fig. 2a) to demonstrate larger field of view and indicated the scale. As it is shown now in this image, the densification intimately related to the transition metal cation migration occurs at the distance of > 15 nm away from the surface that is much more extended than just near-surface reconstruction in Li-rich NMC charged to 4.6 V only (Fig. 2b). Unfortunately, the physics of electron scattering precludes observations of more “bulky” areas due to growing thickness deteriorating the quality of the HAADF-STEM images. The longest distance from the surface to the bulk which was reachable for HAADF-STEM imaging was ~ 20 nm and the M cation migration was clearly observed there as well (see the image at the right). However, taking into account the size of the primary particles of 30-50 nm, the obtained information can be considered essentially “bulk”.

2. A whole-paragraph discussion in Page 7 is based on Fig. S4 and Fig. S5. The quality of the STEM images presented is very poor. The contrast here almost makes it impossible to follow the discussion on TM occupancy. What is the particle size of the pristine material? The bulk information presented in Fig. S4c is at most 10 nm.

Response: It is pity that the referee did not specify his criticism regarding the quality of the HAADF-STEM images as it might appear deteriorated because of various extrinsic factors (brightness/contrast settings of the display, corruption while converting to PDF etc). Anyway, we did our best to provide more clear

images in the revised version by filtering out the experimental noise and adjusting the contrast to facilitate optimal perception. We also placed guidelines to the figures and figure captions to outline the features mentioned in the discussion and make the interpretation easier for the reader. We haven't analyzed the particle size of the pristine material in this work, since a very detailed SEM investigation is provided in a previous work from our group (Chem. Mater. 2017, 29, 23, 9923-9936). The pristine material used in this work was synthesized following the same procedure with the Li/M ratio being 1.35. The pristine powder is made of spherical agglomerates that are

consisting of thin rods propagating from the center of the sphere toward its surface. The thickness of the rods is about ~ 100 -200 nm, and the length could reach whole radius of the agglomerate (i.e., 3-4 μm) as

demonstrated with the BF-STEM image at the right). The rods are composed of the rhombus-like nanocrystals with the size of ~ 30 -50 nm (see Fig. S4 of Supporting information).

3. The authors discussed the effect of OCV at high voltage (4.8 V), it is important to compare the electrochemistry of this material at various testing protocols. From Fig. S16, OCV or CV at 4.8 V results in a significant capacity drop and some voltage decay. What is the origin of such key difference? What is the correlation with phase A and A'?

Response: We thank the reviewer for raising this interesting point that we had not clarified originally. Indeed, resting the material at open circuit voltage (OCV) or applying constant voltage (CV) holding at 4.8 V results in capacity drop and voltage decay. From an electrochemical perspective, additional OCV or CV step results in the reduction of transition metal ions (Co and Ni), as revealed by *in-situ* X-ray absorption spectroscopy measurements, directly contributing to the voltage decay (**we have added a Fig.19 in supplementary information**). The observed capacity drop after imposing the OCV or CV step is closely associated with the gradual phase A \rightarrow A' transition. More specifically, the contraction of TM layers and the growing Mn occupation in the octahedral Li sites upon A \rightarrow A' phase transition certainly would cause a kinetic hindrance in Li⁺ intercalation, leading to the capacity drop. In addition, the formation of phase A' is accompanied with O_{2(gas)} release, which are known to induce particle cracking that could also contribute to the performance deterioration. Concerning the correlation of phase A and A', certainly phase A is the parent phase of phase A'. Upon further delithiation of phase A, the removal of Li⁺ and O create vacancies and destabilize its structure, resulting in structural reorganization (lattice densification) to form phase A'.

4. For comparison, what is the STEM of phase A at the end of 4.6 V charge? Do the authors anticipate the contrasting observation between phase A and phase A' (both bulk and surface) in comparison of 4.6 CC and 4.8 CC-CV?

Response: The ED patterns and HAADF-STEM images of the phase A charged to 4.6 V have been taken as requested by the referee and the contrasting behavior is now shown in Fig. 2 and Fig. S4 and discussed in the text and figure captions.

5. The authors claimed phase A' becomes reversible after the first charge. In Fig. S10, does Mn migration only remain at the surface or in the bulk as well? If in the bulk, the Mn cation trapped in the interlayer octahedral sites in Figure S10 should be highlighted to guide the readers.

Response: After discharge to 2.0 V, synchrotron X-ray diffraction refinement indicates that Mn remains trapped in the interlayer octahedral sites. The overall picture of the M cation migration is now better outlined in Fig. 2 and highlighted in the inserts in this Figure.

Reviewer #3:

1. Throughout the paper the A' phase is referred to as a "densified phase". While this may be true, no clear evidence is provided in the paper. The arguments used to conclude that the A' phase is "densified" are not convincing or logical (bottom of page 6). As I understand the argument, it is concluded that A' is densified since its unit cell volume is smaller than that of the pristine phase. Most layered intercalation compounds contract at high states of charge (certainly along their c-axis), and this occurs without any densification (i.e. transition metal cations filling vacant sites and oxygen vacancies diffusing to the surface). More convincing evidence would be based on a careful refinement of site occupancies. If the authors insist on referring to the A' phase as a "densified phase" throughout the paper (as is the case in the current version), they need to provide stronger evidence for this statement.

Response: We agree with the reviewer that most layered intercalation compounds contract at high states of charge along their c-axis and have experienced this point ourselves in the past when we isolate the CoO₂ phase from electrochemical delithiation of LiCoO₂ (J. Electrochem. Soc. 1996, 143, 3, 1114-1123). Moreover, he/she rightly points out that the origins of such lattice contraction vary from material to material and strongly depends upon the nature of the alkali guests (Li⁺ vs. Na⁺) and on the presence of transition metal occupancy in the van der Waals gap screening the O-O repulsive interactions between layers. **Certainly**, we didn't draw the conclusion of 'densified phase' based on the decrease of c lattice parameter, but based on combined synchrotron and neutron refinement, along with STEM analysis. We are confident that this is bulk. Indeed for X-ray, we used a small wavelength of 0.41 Å and the measurement is done in Debye-Scherrer geometry (transmission through a capillary), the powder pattern we obtained therefore comes from the bulk. For neutrons, it is even more true since neutrons penetrate the whole sample, and the measurement was done on a large volume sample (400 mgs) which was especially prepared for neutrons using a large electrochemical cell.

We use the fact that the c lattice parameter decreases to a value even smaller than the pristine phase as additional supporting experimental evidence. We would also like to recall that, the c lattice parameter of Li-rich NMCs rarely decreases to a value even smaller than the pristine phase upon delithiation. One exception is reported for Li[Ni_{0.2}Li_{0.2}Mn_{0.6}]O₂ by Armstrong *et al* (J. Am. Chem. Soc. 2006, 128, 26, 8694-8698) in which lattice densification was also proposed based on the refinement of neutron powder diffraction.

2. The twinning at high states of charge is interesting and likely of importance to better understanding structural changes that occur at high states of charge. However, the paper is not clear in its description of the twinned structure. For example, it refers to "the mirror-twinned

rhombohedral structure with the (001) twin plane.” However it is not clear whether the (001) indices are for a rhombohedral unit cell, or for the hexagonal unit cell of the O3 crystal. Can the authors consider adding a schematic that illustrates the crystallography of the twins and how they may relate to the original A phase?

Response: Now the atomic model of the twin plane is provided in Fig. S5 along with brief explanations.

3. In the refinement of the A' phase, was tetrahedral Mn occupancy considered? Very few details are provided about the refinements. Are the authors able to speculate on the oxidation state of the tetrahedral Mn in the surface layer? It is not clear why the authors rule out tetrahedral Mn in the bulk phase. It seems that similar features that are attributed to tetrahedral Mn at the surface are also apparent in the bulk (Figure 1f and S4).

Response: Yes, models with Mn in tetrahedral sites were considered for refining the powder patterns of the bulk A' phase, and better refinements were obtained for Mn migration in octahedral sites. An illustration can be seen in figure at right, for which we constructed two models: one with Mn migration in tetrahedral sites (Wyckof position 6c with z close to 0.36) (TOP), and one with the same amount of Mn migration to octahedral sites 3a (i.e., at the same position as Li in Li layers for the pristine material) (BOTTOM). The effect on the synchrotron refinement is shown next to the model, with a zoom on peaks (101), (012)/(006) and (104) as inset. The model with Mn in tetrahedral sites presents a refinement which is worsen compared to the one be obtained with Mn in Li octahedral sites.

Difference Fourier Maps generated from neutron (a) and X-ray (b) diffraction patterns of the A' phase, calculated after having removed all transition metals from the structure (Ni, Mn and Co). On (c) is given a view of the structure as deduced from the combined refinement of the neutron and XRD patterns (oxygen is red, Mn is purple, Li is green, Ni is gray, Co is dark blue)

A second approach was used to ensure that we do not, in the bulk phase, observe massive and long-range ordered migration to tetrahedral sites: from the combined neutron/synchrotron refinement of the A' phase, we removed all transition metals and calculated difference Fourier maps (the figure at right). They clearly show density on the octahedral sites, and not on the tetrahedral sites.

This conclusion is corroborated with the HAADF-STEM image taken at 20 nm away from the surface, which shows the migrated transition metal cations M at the octahedral sites only:

Thus, the migration of the transition metal cations to the tetrahedral sites belongs merely to the near-surface region.

4. The HAADF-STEM images (e.g. S4) appear to show void formation, similar to that observed by Yan *et al* Nature Nanotechnology 602 (2019). Can the authors comment on this (i.e. confirm or rule out that voids have in fact emerged within the bulk of the A' phase)?

Response: To be precise, we have reported the formation of voids in the cycled Li-rich NMC two years before Yan *et al* Nature Nanotechnology 602 (2019) (see Fig. S9 in Chem. Mater. 2017, 29, 23, 9923-9936). These voids are formed only after prolonged electrochemical cycling (ca 100 cycles). In our particular case this uneven contrast has been observed only in the highly charged state (4.8 V) and not at 4.6 V or after discharge to 2.0 V. Thus we attribute it to amorphous layer with uneven thickness which might be part of the cathode/electrolyte interphase (CEI). Investigation of this layer could be an interesting albeit very challenging task, but it goes beyond the scope of this manuscript.

5. Can the authors comment on the recent study by Taylor *et al* JACS, 141, 7333 (2019), which also shows recovered capacity at very low voltage (~ 1.7 V). In this study though, there was clear evidence of the reduction of peroxy-molecules. Could something similar be occurring in Li-excess Mn containing compounds?

Response: We do not think the reduction of peroxy species is occurring in the Li-rich NMC case, since there is no sign of peroxy species ($d_{O-O} \approx 1.45 \text{ \AA}$) formation even at the end of CC - 8 h CV charge as the shortest O-O bond in this structure is 2.644(5) \AA based on the refinement of neutron powder diffraction. However, a possibility could be the possible reduction of oxidized oxygen species ($(O_2)^{n-}$, $4 > n > 2$) at low voltage discharge aside from the reduction of transition metal ions (Mn and Co). We have checked this point by hard X-ray photoelectron spectroscopy and found that by 2 V the 530.5 eV binding energy corresponding to $(O_2)^{n-}$ species was totally absent.

6. The authors measure O₂ release and provide quantitative values. Can the authors translate these numbers into an estimate of how much the crystals have densified?

Response: Indeed, densification of the material is accompanied with O_{2(gas)} release, the fraction of densification may be roughly estimated from the fraction of total O loss. Based on quantitative *on-line* electrochemical mass spectroscopy measurement, 0.15 O is lost from the phase A', i.e., the phase formed at the end of CC - 8 h CV charge, giving 7.5 % total O loss. However, caution must be exercised to implement this estimation, since O loss from the material associate with both bulk densification and surface modifications. Therefore, the estimated fraction of densification would be smaller than 7.5 %. Alternatively, the fraction of densification may be estimated from the decrease of unit cell volume as compared to the pristine phase, $X_{\text{densification}} = (V_{\text{pristine}} - V_{\text{phase A'}})/V_{\text{pristine}}$, i.e., ~ 4.2 % based on the

refinement of neutron powder diffraction, or $\sim 2.8\%$ based on the refinement of synchrotron X-ray diffraction.

Reviewers' comments:

Reviewer #2 (Remarks to the Author):

Regarding the discussion of new phase A' in bulk or surface phase, there are no doubts about the bulk approach of XRD/ND. The challenge of STEM is well known and typically used as a complimentary tool. Nevertheless, the key concern in this discussion is rather the assignment of the final phase after OCV resting. Other supporting experimental evidence is clear. For example, phase A' does not form at 4.6 V, regardless of CV or OCV steps, on the contrary, it forms at 4.8 V-OCV. The criticalness of electrochemical history sounds consistent with the OEMS results about O₂ gas release. New STEM images provided in Fig.2 are much more meaningful compared to that (previous Fig. 1) with a few nanometers on the surface.

The question why it forms after OCV resting will provide scientific advance because bulk densification in the stoichiometric layered R3m does not occur. The authors discussed the possible reasons that explain why you think a densified phase is more likely in this case. Is there reference to support this claim? The DFT calculations maybe helpful for this discussion. I think the diffusion of oxygen vacancies and threshold of oxygen vacancies to stabilize the densified phase A's will be meaningful in this topic. Oxygen vacancies must form in layered-layered oxide due to the oxygen loss, at which point it essentially causes a stable but detrimental densified phase A's will be of interest.

Also, oxygen loss/vacancies is the driving factor to form densified phase A', which is reversible during charge-discharge. However, layered-layered oxide only release O₂ gas in the first cycle, as the authors report here as well. Can the authors comment whether the material is cycled between discharged state and phase A' after the first cycles? In other words, is phase A' the structure at charged state after its formation?

Reviewer #3 (Remarks to the Author):

I have carefully reread the paper and the authors responses. They have addressed my original concerns in the revised manuscript. The paper is in my opinion suitable for publication in Nature Communications.

Below, we have answered **Referee#2's** clarification points by first recalling the question in bold and italic and our response in black while using red to highlight parts that were changed in the revised manuscript.

Reviewer #2

Question 1: The question why it forms after OCV resting will provide scientific advance because bulk densification in the stoichiometric layered R3m does not occur. The authors discussed the possible reasons that explain why you think a densified phase is more likely in this case. Is there reference to support this claim? The DFT calculations maybe helpful for this discussion. I think the diffusion of oxygen vacancies and threshold of oxygen vacancies to stabilize the densified phase A's will be meaningful in this topic. Oxygen vacancies must form in layered-layered oxide due to the oxygen loss, at which point it essentially causes a stable but detrimental densified phase A's will be of interest.

Response: Thanks first for asking this clarification. Indeed, the reviewer is right by pointing out that the release of oxygen and therefore the diffusion of oxygen vacancies is critical for the formation of phase A'. More precisely, upon O₂ release the O/M ratio decreases (*Nature Materials* 2019 18, 496-502) leading to a densification, which is always favorable from a thermodynamic point of view (see <https://materialsproject.org/>). Thus, once oxygen has been released, the phase evolves towards the thermodynamically stable densified phase with a slow kinetics associated to the migration of oxygen vacancies, hence the reason why we see phase still forming after OCV resting. Regarding references to support our discussion of 'bulk densification' in Li-rich NMC, lattice densification in Li[Ni_{0.2}Li_{0.2}Mn_{0.6}]O₂ was also proposed by Armstrong *et al* based on the refinement of neutron powder diffraction (*J. Am. Chem. Soc.* 2006, 128, 26, 8694-8698).

We understand that the Referee #2 suggests DFT calculations to rationalize formation and diffusion of oxygen vacancies. We initially were optimistic in doing such calculations, but they turns out to be quite complex because they first require i) the exact nature of the migrating species (O⁻, O²⁻, O₂ⁿ⁻ etc), their coupling with the transition metal cation migration, possible clustering of the O vacancies and cation vacancies etc and ii) the knowledge of the

diffusion paths to go from one phase to another, hence necessitating an accurate positioning of the atoms within the A' phase. Classical molecular dynamics (MD) (much more affordable than ab initio MD) could surely make it possible to work on larger systems and to follow structural changes at the nanoscopic scale, but classical MD is not suitable for transition metals for which accurate force fields still need to be developed. Moreover, to simulate bonds formation/breaking, reactive force fields are required which constitutes today one of the biggest challenge for classical MD methods (Computational Materials volume 2, Article number: 15011 (2016)). Hence, the reason why we gave up on our DFT-MD calculations attempts.

Therefore, to reflect the referee questions we add this paragraph in the discussion section:

'...Let's recall that the densification associated to a decrease of the O/M ratio is always favorable from a thermodynamic point of view (*Nature Materials* 2019 18, 496-502), thus its appearance after either OCV resting or CV holding is indicative of a kinetically limited growth of the thermodynamically stable densified phase A', most likely due to a sluggish migration of oxygen vacancies. Although there are indications that the energy barriers for the O vacancy migration in Li-rich layered oxides are lower than those in the stoichiometric LiMO₂ phases (*Adv. Energy Mater.* 2018 9, 1802586), these barriers can hardly be computed precisely because of too many unknowns, such as the exact nature of the migrating species (O⁻, O²⁻, O₂ⁿ⁻ etc), possible clustering of the O vacancies and cation vacancies and multiple possibilities for the diffusion pathways.'

Question 2: Also, oxygen loss/vacancies is the driving factor to form densified phase A', which is reversible during charge-discharge. However, layered-layered oxide only release O₂ gas in the first cycle, as the authors report here as well. Can the authors comment whether the material is cycled between discharged state and phase A' after the first cycles? In other words, is phase A' the structure at charged state after its formation?

Response: Thanks again for the reviewer for this well spotted question that we have considered but did not bother to extend on the paper. Phase A' is the structure at charged state after its formation. As shown by *operando* XRD, phase A' repeatedly forms at the end of each CC-CV charge until the 26th cycle (the maximum number of cycles we have tried) (Fig. 3a). A more detailed explanation can be found in page 9 of our revised manuscript:

'...We further tracked down both the structural evolution and O₂ release upon subsequent cycling. From *operando* XRD, we could deduce that the phase A' repeatedly forms at the end of each CC-CV charge until the 26th cycle (the maximum number of cycles we have tried) (Fig. 3a), whereas no O₂ release was observed in the 2nd cycle (Fig. 3b). Altogether, these results imply that during the initial charge, the formation of phase A', whose existence ranges from x = 0.15 to 0.05 (see dashed square in Fig. 3a), is caused by the conjoint removals of Li⁺ and oxygen that creates both cationic and anionic vacancies followed by lattice densification. Once it forms, the new phase A' can reversibly uptake and release Li⁺ without the need for having more anionic vacancies hence O₂ release.'